# INTERPRETING KNOWLEDGE GRAPH RELATION REPRESENTATION FROM WORD EMBEDDINGS

**Carl Allen**[1*]**, Ivana Balažević**[1*] **& Timothy Hospedales**[1,2]
[1] University of Edinburgh, UK      [2] Samsung AI Centre, Cambridge, UK
{carl.allen, ivana.balazevic, t.hospedales}@ed.ac.uk

## ABSTRACT

Many models learn representations of knowledge graph data by exploiting its low-rank latent structure, encoding known relations between entities and enabling unknown facts to be inferred. To predict whether a relation holds between entities, embeddings are typically compared in the latent space following a relation-specific mapping. Whilst their predictive performance has steadily improved, how such models capture the underlying latent structure of semantic information remains unexplained. Building on recent theoretical understanding of word embeddings, we categorise knowledge graph relations into three types and for each derive explicit requirements of their representations. We show that empirical properties of relation representations and the relative performance of leading knowledge graph representation methods are justified by our analysis.

## 1 INTRODUCTION

Knowledge graphs are large repositories of binary relations between words (or entities) in the form of *(subject, relation, object)* triples. Many models for representing entities and relations have been developed, so that known facts can be recalled and previously unknown facts can be inferred, a task known as *link prediction*. Recent link prediction models (e.g. Bordes et al., 2013; Trouillon et al., 2016; Balažević et al., 2019b) learn entity representations, or *embeddings*, of far lower dimensionality than the number of entities, by capturing latent structure in the data. Relations are typically represented as a mapping from the embedding of a subject entity to those of related object entities. Although the performance of link prediction models has steadily improved for nearly a decade, relatively little is understood of the low-rank latent structure that underpins them, which we address in this work. The outcomes of our analysis can be used to aid and direct future knowledge graph model design.

We start by drawing a parallel between the entity embeddings of knowledge graphs and context-free word embeddings, e.g. as learned by Word2Vec (W2V) (Mikolov et al., 2013a) and GloVe (Pennington et al., 2014). Our motivating premise is that the same latent word features (e.g. meaning(s), tense, grammatical type) give rise to the patterns found in different data sources, i.e. manifesting in word co-occurrence statistics and determining which words relate to which. Different embedding approaches may capture such structure in different ways, but if it is fundamentally the same, an understanding gained from one embedding task (e.g. word embedding) may benefit another (e.g. knowledge graph representation). Furthermore, the relatively limited but accurate data used in knowledge graph representation differs materially from the highly abundant but statistically noisy text data used for word embeddings. As such, theoretically reconciling the two embedding methods may lead to unified and improved embeddings learned jointly from both data sources.

Recent work (Allen & Hospedales, 2019; Allen et al., 2019) theoretically explains how semantic properties are encoded in word embeddings that (approximately) factorise a matrix of *pointwise mutual information* (PMI) from word co-occurrence statistics, as known for W2V (Levy & Goldberg, 2014). *Semantic* relationships between words, specifically similarity, relatedness, paraphrase and analogy, are proven to manifest as linear *geometric* relationships between rows of the PMI matrix (subject to known error terms), of which word embeddings can be considered low-rank projections. This explains, for example, the observations that similar words have similar embeddings and that embeddings of analogous word pairs share a common "vector offset" (e.g. Mikolov et al., 2013b).

---

*Equal contribution

**Table 1:** Score functions of representative linear link prediction models. $\boldsymbol{R} \in \mathbb{R}^{d_e \times d_e}$ and $\boldsymbol{r} \in \mathbb{R}^{d_e}$ are the relation matrix and translation vector, $\mathsf{W} \in \mathbb{R}^{d_e \times d_r \times d_e}$ is the core tensor and $b_s, b_o \in \mathbb{R}$ are the entity biases.

| Model | | Linear Subcategory | Score Function |
|---|---|---|---|
| TransE | (Bordes et al., 2013) | additive | $-\|\boldsymbol{e}_s + \boldsymbol{r} - \boldsymbol{e}_o\|_2^2$ |
| DistMult | (Yang et al., 2015) | multiplicative (diagonal) | $\boldsymbol{e}_s^\top \boldsymbol{R} \boldsymbol{e}_o$ |
| TuckER | (Balažević et al., 2019b) | multiplicative | $\mathsf{W} \times_1 \boldsymbol{e}_s \times_2 \boldsymbol{r} \times_3 \boldsymbol{e}_o$ |
| MuRE | (Balažević et al., 2019a) | multiplicative (diagonal) + additive | $-\|\boldsymbol{R}\boldsymbol{e}_s + \boldsymbol{r} - \boldsymbol{e}_o\|_2^2 + b_s + b_o$ |

We extend this insight to identify geometric relationships between PMI-based word embeddings that correspond to other relations, i.e. those of knowledge graphs. Such *relation conditions* define relation-specific mappings between entity embeddings (i.e. *relation representations*) and so provide a "blue-print" for knowledge graph representation models. Analysing the relation representations of leading knowledge graph representation models, we find that various properties, including their relative link prediction performance, accord with predictions based on these relation conditions, supporting the premise that a *common latent structure* is learned by word and knowledge graph embedding models, despite the significant differences between their training data and methodology.

In summary, the key contributions of this work are:
- to use recent understanding of PMI-based word embeddings to derive geometric attributes of a relation representation for it to map subject word embeddings to all related object word embeddings (*relation conditions*), which partition relations into three *types* (§3);
- to show that both per-relation ranking as well as classification performance of leading link prediction models corresponds to the model satisfying the appropriate relation conditions, i.e. how closely its relation representations match the geometric form derived theoretically (§4.1); and
- to show that properties of knowledge graph representation models fit predictions based on relation conditions, e.g. the strength of a relation's *relatedness* aspect is reflected in the eigenvalues of its relation matrix (§4.2).

## 2 BACKGROUND

**Knowledge graph representation:** Recent knowledge graph models typically represent entities $e_s, e_o$ as vectors $\boldsymbol{e}_s, \boldsymbol{e}_o \in \mathbb{R}^{d_e}$, and relations as transformations in the latent space from subject to object entity embedding, where the dimension $d_e$ is far lower (e.g. 200) than the number of entities $n_e$ (e.g. $> 10^4$). Such models are distinguished by their *score function*, which defines (i) the form of the relation transformation, e.g. matrix multiplication and/or vector addition; and (ii) the measure of proximity between a transformed subject embedding and an object embedding, e.g. dot product or Euclidean distance. Score functions can be non-linear (e.g. Dettmers et al., 2018), or linear and sub-categorised as *additive*, *multiplicative* or both. We focus on linear models due to their simplicity and strong performance at link prediction (including state-of-the-art). Table 1 shows the score functions of competitive linear knowledge graph embedding models spanning the sub-categories: TransE (Bordes et al., 2013), DistMult (Yang et al., 2015), TuckER (Balažević et al., 2019b) and MuRE (Balažević et al., 2019a).

*Additive models* apply a relation-specific translation to a subject entity embedding and typically use Euclidean distance to evaluate proximity to object embeddings. A generic additive score function is given by $\phi(e_s, r, e_o) = -\|\boldsymbol{e}_s + \boldsymbol{r} - \boldsymbol{e}_o\|_2^2 + b_s + b_o$. A simple example is TransE, where $b_s = b_o = 0$.

*Multiplicative models* have the generic score function $\phi(e_s, r, e_o) = \boldsymbol{e}_s^\top \boldsymbol{R} \boldsymbol{e}_o$, i.e. a bilinear product of the entity embeddings and a relation-specific matrix $\boldsymbol{R}$. DistMult is a simple example with $\boldsymbol{R}$ diagonal and so cannot model asymmetric relations (Trouillon et al., 2016). In TuckER, each relation-specific $\boldsymbol{R} = \mathsf{W} \times_3 \boldsymbol{r}$ is a linear combination of $d_r$ "prototype" relation matrices in a core tensor $\mathsf{W} \in \mathbb{R}^{d_e \times d_r \times d_e}$ ($\times_n$ denoting tensor product along mode $n$), facilitating *multi-task learning* across relations.

Some models, e.g. MuRE, combine both multiplicative ($\boldsymbol{R}$) and additive ($\boldsymbol{r}$) components.

**Word embedding:** Algorithms such as Word2Vec (Mikolov et al., 2013a) and GloVe (Pennington et al., 2014) generate low-dimensional word embeddings that perform well on downstream tasks (Baroni et al., 2014). Such models predict the context words ($c_j$) observed around a target word ($w_i$) in a text corpus using shallow neural networks. Whilst recent language models (e.g. Devlin et al., 2018; Peters et al., 2018) achieve strong performance using *contextualised* word embeddings, we focus on "context-free" embeddings since knowledge graph entities have no obvious context and, importantly, they offer insight into embedding interpretability.

Levy & Goldberg (2014) show that, for a dictionary of $n_e$ unique words and embedding dimension $d_e \ll n_e$, W2V's loss function is minimised when its embeddings $\boldsymbol{w}_i, \boldsymbol{c}_j$ form matrices $\boldsymbol{W}, \boldsymbol{C} \in \mathbb{R}^{d_e \times n_e}$ that factorise a *pointwise mutual information* (PMI) matrix of word co-occurrence statistics (PMI$(w_i, c_j) = \log \frac{P(w_i, c_j)}{P(w_i)P(c_j)}$), subject to a shift term. This result relates W2V to earlier count-based embeddings and specifically PMI, which has a history in linguistic analysis (Turney & Pantel, 2010). From its loss function, GloVe can be seen to perform a related factorisation.

Recent work (Allen & Hospedales, 2019; Allen et al., 2019) shows how the semantic relationships of *similarity*, *relatedness*, *paraphrase* and *analogy* are encoded in PMI-based word embeddings by recognising such embeddings as low-rank projections of high dimensional rows of the PMI matrix, termed *PMI vectors*. Those semantic relationships are described in terms of *multiplicative* interactions between co-occurrence probabilities (subject to defined error terms), that correspond to *additive* interactions between (logarithmic) PMI statistics, and hence PMI vectors. Thus, under a sufficiently linear projection, those semantic relationships correspond to linear relationships between word embeddings. Note that although the relative geometry reflecting semantic relationships is preserved, the direct interpretability of dimensions, as in PMI vectors, is lost since the embedding matrices can be arbitrarily scaled/rotated if the other is inversely transformed. We state the relevant semantic relationships on which we build, denoting the set of unique dictionary words by $\mathcal{E}$:

- **Paraphrase**: word subsets $\mathcal{W}, \mathcal{W}^* \subseteq \mathcal{E}$ are said to *paraphrase* if they induce similar distributions over nearby words, i.e. $p(\mathcal{E}|\mathcal{W}) \approx p(\mathcal{E}|\mathcal{W}^*)$, e.g. {*king*} paraphrases {*man, royal*}.
- **Analogy**: a common example of an *analogy* is "*woman* is to *queen* as *man* is to *king*" and can be defined as any set of word pairs $\{(w_i, w_i^*)\}_{i \in \mathcal{I}}$ for which it is semantically meaningful to say "$w_a$ is to $w_a^*$ as $w_b$ is to $w_b^*$" $\forall a, b \in \mathcal{I}$.

Where one word subset paraphrases another, the sums of their embeddings are shown to be equal (subject to the independence of words within each set), e.g. $\boldsymbol{w}_{king} \approx \boldsymbol{w}_{man} + \boldsymbol{w}_{royal}$. An interesting connection is established between the two semantic relationships: a set of word pairs $\mathcal{A} = \{(w_a, w_a^*), (w_b, w_b^*)\}$ is an analogy if $\{w_a, w_b^*\}$ paraphrases $\{w_a^*, w_b\}$, in which case the embeddings satisfy $\boldsymbol{w}_{a^*} - \boldsymbol{w}_a \approx \boldsymbol{w}_{b^*} - \boldsymbol{w}_b$ ("vector offset").

## 3  FROM ANALOGIES TO KNOWLEDGE GRAPH RELATIONS

Analogies from the field of word embeddings are our starting point for developing a theoretical basis for representing knowledge graph relations. The relevance of analogies stems from the observation that for an analogy to hold (see §2), its word pairs, e.g {*(man, king), (woman, queen), (girl, princess)*}, must be *related* in the same way, comparably to subject-object entity pairs under a common knowledge graph relation. Our aim is to develop the understanding of PMI-based word embeddings (henceforth *word embeddings*), to identify the mathematical properties necessary for a relation representation to map subject word embeddings to all related object word embeddings.

Considering the paraphrasing word sets {*king*} and {*man, royal*} corresponding to the word embedding relationship $\boldsymbol{w}_{king} \approx \boldsymbol{w}_{man} + \boldsymbol{w}_{royal}$ (§2), *royal* can be interpreted as the semantic difference between *man* and *king*, fitting intuitively with the relationship $\boldsymbol{w}_{royal} \approx \boldsymbol{w}_{king} - \boldsymbol{w}_{man}$. Fundamentally, this relationship holds because the difference between words that co-occur (i.e. occur more frequently than if independent) with *king* and those that co-occur with *man*, reflects those words that co-occur with *royal*. We refer to this difference in co-occurrence distribution as a "context shift", from *man* (subject) to *king* (object). Allen & Hospedales (2019) effectively show that where multiple word pairs share a common context shift, they form an analogy whose embeddings satisfy the vector offset relationship. This result seems obvious where the context shift mirrors an identifiable word, the embedding of which is approximated by the common vector offset, e.g. *queen* and *woman* are related by the same context shift, i.e. $\boldsymbol{w}_{queen} \approx \boldsymbol{w}_{woman} + \boldsymbol{w}_{royal}$, thus $\boldsymbol{w}_{queen} - \boldsymbol{w}_{woman} \approx \boldsymbol{w}_{king} - \boldsymbol{w}_{man}$. However, the same result holds, i.e. an analogy is formed with a common vector offset between embeddings, for an arbitrary (common) context shift that may reflect no particular word. Importantly, these context shift relations evidence a case in which it is known how a relation can be represented, i.e. by an additive vector (comparable to TransE) *if* entities are represented by word embeddings. More generally, this provides an interpretable foothold into relation representation.

Note that not all sets of word pairs considered analogies exhibit a clear context shift relation, e.g. in the analogy {*(car,engine), (bus,seats)*}, the difference between words co-occurring with *engine* and *car* is not expected to reflect the corresponding difference between *bus* and *seats*. This illustrates how

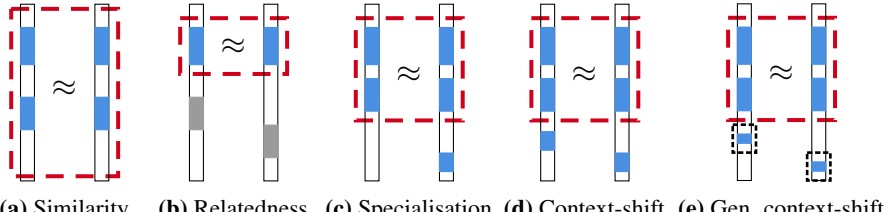

**(a)** Similarity  **(b)** Relatedness  **(c)** Specialisation  **(d)** Context-shift  **(e)** Gen. context-shift

**Figure 1:** Relationships between PMI vectors (black rectangles) of subject/object words for different relation *types*. PMI vectors capture co-occurrence with every dictionary word: strong associations (PMI > 0) are shaded (blue define the relation, grey are random other associations); red dash = *relatedness*; black dash = *context sets*.

analogies are a loosely defined concept, e.g. their implicit relation may be semantic or syntactic, with several sub-categories of each (e.g. see Gladkova et al. (2016)). The same is readily observed for the relations of knowledge graphs. This likely explains the observed variability in "solving" analogies by use of vector offset (e.g. Köper et al., 2015; Karpinska et al., 2018; Gladkova et al., 2016) and suggests that further consideration is required to represent relations (or solve analogies) in general.

We have seen that the existence of a context shift relation between a subject and object word implies a (relation-specific) geometric relationship between word embeddings, thus the latter provides a *necessary condition for the relation to hold*. We refer to this as a "relation condition" and aim to identify relation conditions for other classes of relation. Once identified, relation conditions define a mapping from subject embeddings to all related object embeddings, by which related entities might be identified with a proximity measure (e.g. Euclidean distance or dot product). This is the precise aim of a knowledge graph representation model, but loss functions are typically developed heuristically. Given the existence of many representation models, we can verify identified relation conditions by contrasting the per-relation performance of various models with the extent to which their loss function reflects the appropriate relation conditions. Note that since relation conditions are necessary rather than sufficient, they do not guarantee a relation holds, i.e. false positives may arise.

Whilst we seek to establish relation conditions based on PMI word embeddings, the data used to train knowledge graph embeddings differs significantly to the text data used by word embeddings, and the relevance of conditions ultimately based on PMI statistics may seem questionable. However, where a knowledge graph representation model implements relation conditions and measures proximity between embeddings, the parameters of word embeddings necessarily provide *a potential* solution that minimises the loss function. Many equivalent solutions may exist due to symmetry as typical for neural network architectures. We now define relation types and identify their relation conditions (underlined); we then consider the completeness of this categorisation.

• **Similarity:** Semantically similar words induce similar distributions over the words they co-occur with. Thus their PMI vectors and word embeddings are similar (Fig 1a).

• **Relatedness:** The relatedness of two words can be considered in terms of the words $\mathcal{S} \subseteq \mathcal{E}$ with which both co-occur similarly. $\mathcal{S}$ defines the *nature* of relatedness, e.g. *milk* and *cheese* are related by $\mathcal{S} = \{dairy, breakfast, ...\}$; and $|\mathcal{S}|$ reflects the *strength* of relatedness. Since PMI vector components corresponding to $\mathcal{S}$ are similar (Fig 1b), embeddings of $\mathcal{S}$-*related* words have similar components in the subspace $\mathbb{V}_{\mathcal{S}}$ that spans the projected PMI vector dimensions corresponding to $\mathcal{S}$. The rank of $\mathbb{V}_{\mathcal{S}}$ is thus anticipated to reflect relatedness strength. Relatedness can be seen as a weaker and more variable generalisation of similarity, its limiting case where $\mathcal{S} = \mathcal{E}$, hence rank$(\mathbb{V}_{\mathcal{S}}) = d_e$.

• **Context-shift:** As discussed above, words related by a common difference between their distributions of co-occurring words, defined as *context-shifts*, share a common vector offset between word embeddings. Context might be considered *added* (e.g. *man* to *king*), termed a **specialisation** (Fig 1c), *subtracted* (e.g. *king* to *man*) or both (Fig 1d). These relations are 1-to-1 (subject to synonyms) and include an aspect of *relatedness* due to the word associations in common. Note that, specialisations include hyponyms/hypernyms and context shifts include meronyms.

• **Generalised context-shift:** Context-shift relations generalise to 1-to-many, many-to-1 and many-to-many relations where the added/subtracted context may be from a (relation-specific) *context set* (Fig 1e), e.g. *any* city or *anything* bigger. The potential scope and size of context sets adds variability to these relations. The limiting case in which the context set is "small" reduces to a 1-to-1 context-shift (above) and the embedding difference is a known vector offset. In the limiting case of a "large" context set, the added/subtracted context is essentially unrestricted such that only the relatedness aspect of the relation, and thus a common subspace component of embeddings, is fixed.

**Categorisation completeness:** Taking intuition from Fig 1 and considering PMI vectors as *sets of word features*, these relation types can be interpreted as set operations: similarity as set equality; relatedness as subset equality; and context-shift as a relation-specific set difference. Since for any relation each feature must either remain unchanged (relatedness), change (context shift) or else be irrelevant, we conjecture that the above relation types give a complete partition of semantic relations.

**Table 2:** Categorisation of WN18RR relations.

| Type | Relation | Examples *(subject entity, object entity)* |
|------|----------|---------------------------------------------|
| R | verb_group
derivationally_related_form
also_see | *(trim_down_VB_1, cut_VB_35), (hatch_VB_1, incubate_VB_2)*
*(lodge_VB_4, accommodation_NN_4), (question_NN_1, inquire_VB_1)*
*(clean_JJ_1, tidy_JJ_1), (ram_VB_2, screw_VB_3)* |
| S | hypernym
instance_hypernym | *(land_reform_NN_1, reform_NN_1), (prickle-weed_NN_1, herbaceous_plant_NN_1)*
*(yellowstone_river_NN_1, river_NN_1), (leipzig_NN_1, urban_center_NN_1)* |
| C | member_of_domain_usage
member_of_domain_region
member_meronym
has_part
synset_domain_topic_of | *(colloquialism_NN_1, figure_VB_5), (plural_form_NN_1, authority_NN_2)*
*(rome_NN_1, gladiator_NN_1), (usa_NN_1, multiple_voting_NN_1)*
*(south_NN_2, sunshine_state_NN_1), (genus_carya_NN_1, pecan_tree_NN_1)*
*(aircraft_NN_1, cabin_NN_3), (morocco_NN_1, atlas_mountains_NN_1)*
*(quark_NN_1, physics_NN_1), (harmonize_VB_3, music_NN_4)* |

### 3.1 Categorising Real Knowledge Graph Relations

Analysing the relations of popular knowledge graph datasets, we observe that they indeed imply (i) a relatedness aspect reflecting a common theme (e.g. both entities are animals or geographic terms); and (ii) contextual themes specific to the subject and/or object entities. Further, relations fall under a hierarchy of three *relation types*: highly related (**R**); generalised specialisation (**S**); and generalised context-shift (**C**). As above, "generalised" indicates that context differences are not restricted to be 1-1. From Fig 1, it can be seen that type R relations are a special case of S, which are a special case of C. Thus type C encompasses all considered relations. Whilst there are many ways to classify relations, e.g. by hierarchy, transitivity, the proposed relation conditions delineate relations by the required mathematical form (and complexity) of their representation. Table 2 shows a categorisation of the relations of the WN18RR dataset (Dettmers et al., 2018) comprising 11 relations and 40,943 entities.[1] An explanation for the category assignment is in Appx. A. Analysing the commonly used FB15k-237 dataset (Toutanova et al., 2015) reveals relations to be almost exclusively of type C, precluding a contrast of performance per relation type and hence that dataset is omitted from our analysis. Instead, we categorise a random subsample of 12 relations from the NELL-995 dataset (Xiong et al., 2017), containing 75,492 entities and 200 relations (see Tables 8 and 9 in Appx. B).

### 3.2 Relations as mappings between embeddings

Given the relation conditions of a relation type, we now consider mappings that satisfy them and thereby loss functions able to identify relations of each type, evaluating proximity between mapped entity embeddings by dot product or Euclidean distance. We then contrast our theoretically derived loss functions, specific to a relation type, with those of several knowledge graph models (Table 1) to predict identifiable properties and the relative performance of different knowledge graph models for each relation type.

**R:** Identifying $\mathcal{S}$-relatedness requires testing both entity embeddings $e_s, e_o$ for a common subspace component $\mathbb{V}_{\mathcal{S}}$, which can be achieved by projecting both embeddings onto $\mathbb{V}_{\mathcal{S}}$ and comparing their images. Projection requires multiplication by a matrix $P_r \in \mathbb{R}^{d \times d}$ and cannot be achieved additively, except in the trivial limiting case of similarity ($P_r = I$) when $r \approx 0$ can be added.

Comparison by dot product gives $(P_r e_s)^\top (P_r e_o) = e_s^\top P_r^\top P_r e_o = e_s^\top M_r e_o$ (for relation-specific symmetric $M_r = P_r^\top P_r$). Euclidean distance gives $\|P_r e_s - P_r e_o\|^2 = (e_s - e_o)^\top M_r (e_s - e_o) = \|P_r e_s\|^2 - 2 e_s^\top M_r e_o + \|P_r e_o\|^2$.

**S/C:** The relation conditions require testing for both $\mathcal{S}$-relatedness and relation-specific entity component(s) $(v_r^s, v_r^o)$. This is achieved by (i) multiplying both entity embeddings by a relation-specific projection matrix $P_r$ that projects onto the subspace that spans the low-rank projection of dimensions corresponding to $\mathcal{S}$, $v_r^s$ and $v_r^o$ (i.e. testing for $\mathcal{S}$-relatedness while preserving relation-specific entity components); and (ii) adding a relation-specific vector $r = v_r^o - v_r^s$ to the transformed subject entity embeddings.

---

[1] We omit the relation "similar_to" since its instances have no discernible structure, and only 3 occur in the test set, all of which are the inverse of a training example and trivial to predict.

Comparing the transformed entity embeddings by dot product equates to $(\boldsymbol{P}_r\boldsymbol{e}_s + \boldsymbol{r})^\top \boldsymbol{P}_r\boldsymbol{e}_o$; and by Euclidean distance to $\|\boldsymbol{P}_r\boldsymbol{e}_s + \boldsymbol{r} - \boldsymbol{P}_r\boldsymbol{e}_o\|^2 = \|\boldsymbol{P}_r\boldsymbol{e}_s + \boldsymbol{r}\|^2 - 2(\boldsymbol{P}_r\boldsymbol{e}_s + \boldsymbol{r})^\top \boldsymbol{P}_r\boldsymbol{e}_o + \|\boldsymbol{P}_r\boldsymbol{e}_o\|^2$ (*cf* MuRE: $\|\boldsymbol{R}\boldsymbol{e}_s + \boldsymbol{r} - \boldsymbol{e}_o\|^2$).

Contrasting these theoretically derived loss functions with those of knowledge graph models (Table 1), we make the following predictions:

**P1:** The ability to learn the representation of a relation is expected to reflect:
   (a) the complexity of its type (R<S<C) independently of model choice; and
   (b) whether relation conditions (e.g. additive/multiplicative interactions) are met by the model.

**P2:** Knowledge graph relation representations reflect the following type-specific properties:
   (a) relation matrices for relatedness (type R) relations are highly symmetric;
   (b) offset vectors for relatedness relations have low norm; and
   (c) as a proxy to the rank of $\mathbb{V}_\mathcal{S}$, the eigenvalues of a relation matrix reflect relatedness strength.

To elaborate, our core prediction P1(b) anticipates that: (i) additive-only models (e.g. TransE) are not suited to identifying the relatedness aspect of relations, except in limiting cases of similarity, requiring a zero vector); (ii) multiplicative-only models (e.g. DistMult) should perform well on type R relations, but are not suited to identifying entity-specific features of type S/C (an asymmetric relation matrix in TuckER may help compensate); and (iii) the loss function of MuRE closely resembles that derived for type C relations, which generalise all others, and is thus expected to perform best overall.

## 4 Evidence linking knowledge graph and word embeddings

We test whether the predictions P1 and P2, made on the basis of word embeddings, apply to knowledge graph relations by analysing the performance and properties of competitive knowledge graph models. We compare TransE, DistMult, TuckER and MuRE, which entail different forms of relation representation, on all WN18RR relations and a similar number of NELL-995 relations (spanning all relation types). All models have a comparable number of free parameters.

Since for TransE, the logistic sigmoid cannot be applied to the score function to give a probabilistic interpretation comparable to other models, for fair comparison we include MuRE$_I$, a constrained variant of MuRE with $\boldsymbol{R}_s = \boldsymbol{R}_o = \boldsymbol{I}$, as a proxy to TransE. Implementation details are included in Appx. D. For evaluation, we generate $2n_e$ *evaluation triples* for each test triple ($n_e = |\mathcal{E}|$ denoting the number of entities) by fixing the subject entity $e_s$ and relation $r$ and replacing the object entity $e_o$ with each entity in turn and then keeping $e_o$ and $r$ fixed and varying $e_s$. Each model's scores for the evaluation triples are ranked to give the standard metric Hits@10 (Bordes et al., 2013), i.e. the fraction of times a true triple appears in the top 10 ranked evaluation triples.

### 4.1 P1: Justifying the relative performance of knowledge graph models

**Ranking performance:** Tables 3 and 4 report Hits@10 for each relation and include the relation type as well as known confounding influences: percentage of relation instances in the training and test sets (approximately equal), the actual number of instances in the test set (causing some results to be highly granular), Krackhardt hierarchy score (see Appx. E) (Krackhardt, 2014; Balažević et al., 2019a) and maximum and average shortest path between any two related nodes. A further confounding effect is dependence between relations: Lacroix et al. (2018) and Balažević et al. (2019b) independently show that constraining the rank of relation representations is beneficial for datasets with many relations due to *multi-task learning*, particularly when the number of instances per relation is low. This is expected to benefit TuckER on the NELL-995 dataset (200 relations).

As predicted in P1(a), all models tend to perform best at type R relations, with a clear performance gap to other relation types. Also, performance on type S relations appears higher in general than type C. In accordance with P1(b), additive-only models (TransE, MuRE$_I$) perform worst on average since all relation types involve (multiplicative) relatedness. Best performance is achieved on type R relations, which can be represented by a small/zero additive vector. Multiplicative-only DistMult performs well, sometimes best, on type R relations, fitting expectation as it can represent those relations and has no inessential parameters, e.g. that may overfit to noise, which may explain instances where MuRE performs slightly worse. As expected, MuRE, performs best overall (particularly on WN18RR), and most strongly on S and C type relations, predicted to require both multiplicative and additive components. Comparable performance of TuckER on NELL-995 may be explained by its multi-task learning ability.

**Table 3:** Hits@10 per relation on WN18RR.

| Relation Name | Type | % | # | Khs | Max/Avg Path | | TransE | MuRE$_I$ | DistMult | TuckER | MuRE |
|---|---|---|---|---|---|---|---|---|---|---|---|
| verb_group | R | 1% | 39 | 0.00 | - | - | 0.87 | 0.95 | **0.97** | **0.97** | **0.97** |
| derivationally_related_form | R | 34% | 1074 | 0.04 | - | - | 0.93 | 0.96 | 0.96 | 0.96 | **0.97** |
| also_see | R | 2% | 56 | 0.24 | 44 | 15.2 | 0.59 | **0.73** | 0.67 | 0.72 | **0.73** |
| instance_hypernym | S | 4% | 122 | 1.00 | 3 | 1.0 | 0.22 | 0.52 | 0.47 | 0.53 | **0.54** |
| synset_domain_topic_of | C | 4% | 114 | 0.99 | 3 | 1.1 | 0.19 | 0.43 | 0.42 | 0.45 | **0.53** |
| member_of_domain_usage | C | 1% | 24 | 1.00 | 2 | 1.0 | 0.42 | 0.42 | 0.48 | 0.38 | **0.50** |
| member_of_domain_region | C | 1% | 26 | 1.00 | 2 | 1.0 | 0.35 | 0.40 | 0.40 | 0.35 | **0.46** |
| member_meronym | C | 8% | 253 | 1.00 | 10 | 3.9 | 0.04 | 0.38 | 0.30 | **0.39** | **0.39** |
| has_part | C | 6% | 172 | 1.00 | 13 | 2.2 | 0.04 | 0.31 | 0.28 | 0.29 | **0.35** |
| hypernym | S | 40% | 1251 | 0.99 | 18 | 4.5 | 0.02 | 0.20 | 0.19 | 0.20 | **0.28** |
| all | | 100% | 3134 | | | | 0.38 | 0.52 | 0.51 | 0.53 | **0.57** |

**Table 4:** Hits@10 per relation on NELL-995.

| Relation Name | Type | % | # | Khs | Max/Avg Path | | TransE | MuRE$_I$ | DistMult | TuckER | MuRE |
|---|---|---|---|---|---|---|---|---|---|---|---|
| teamplaysagainstteam | R | 2% | 243 | 0.11 | 10 | 3.5 | 0.76 | 0.84 | **0.90** | 0.89 | 0.89 |
| clothingtogowithclothing | R | 1% | 132 | 0.17 | 5 | 2.6 | 0.72 | 0.80 | **0.88** | 0.85 | 0.84 |
| professionistypeofprofession | S | 1% | 143 | 0.38 | 7 | 2.5 | 0.37 | 0.55 | 0.62 | 0.65 | **0.66** |
| animalistypeofanimal | S | 1% | 103 | 0.68 | 9 | 3.1 | 0.50 | 0.56 | 0.64 | **0.68** | 0.65 |
| athleteplayssport | C | 1% | 113 | 1.00 | 1 | 1.0 | 0.54 | 0.58 | 0.58 | 0.60 | **0.64** |
| chemicalistypeofchemical | S | 1% | 115 | 0.53 | 6 | 3.0 | 0.23 | 0.43 | 0.49 | 0.51 | **0.60** |
| itemfoundinroom | C | 2% | 162 | 1.00 | 1 | 1.0 | 0.39 | 0.57 | 0.53 | 0.56 | **0.59** |
| agentcollaborateswithagent | R | 1% | 119 | 0.51 | 14 | 4.7 | 0.44 | 0.58 | **0.64** | 0.61 | 0.58 |
| bodypartcontainsbodypart | C | 1% | 103 | 0.60 | 7 | 3.2 | 0.30 | 0.38 | 0.54 | **0.58** | 0.55 |
| atdate | C | 10% | 967 | 0.99 | 4 | 1.1 | 0.41 | 0.50 | 0.48 | 0.48 | **0.52** |
| locationlocatedwithinlocation | C | 2% | 157 | 1.00 | 6 | 1.9 | 0.35 | 0.37 | 0.46 | **0.48** | **0.48** |
| atlocation | C | 1% | 294 | 0.99 | 6 | 1.4 | 0.22 | 0.33 | 0.39 | 0.43 | **0.44** |
| all | | 100% | 20000 | | | | 0.36 | 0.48 | 0.51 | **0.52** | **0.52** |

Other anomalous results also closely align with confounding factors. For example, all models perform poorly on the *hypernym* relation, despite it having a relative abundance of training data (40% of all instances), which may be explained by its *hierarchical* nature (Khs ≈ 1 and long paths). The same may explain the reduced performance on relations *also_see* and *agentcollaborateswithagent*. As found previously (Balažević et al., 2019a), none of the models considered are well suited to modelling hierarchical structures. We also note that the percentage of training instances of a relation is not a dominant factor on performance, as would be expected if all relations could be equally represented.

**Classification performance:** We further evaluate whether P1 holds when comparing knowledge graph models by classification accuracy on WN18RR. Independent predictions of whether a given triple is true or false are not commonly evaluated, instead metrics such as mean reciprocal rank and Hits@$k$ are reported that compare the prediction of a test triple against all evaluation triples. Not only is this computationally costly, the evaluation is flawed if an entity is related to $l > k$ others ($k$ is often 1 or 3). A correct prediction validly falling within the top $l$ but not the top $k$ would appear incorrect under the metric. Some recent works also note the importance of standalone predictions (Speranskaya et al., 2020; Pezeshkpour et al., 2020).

Since for each relation there are $n_e^2$ possible entity-entity relationships, we sub-sample by computing predictions only for those $(e_s, r, e_o)$ triples for which the $e_s, r$ pairs appear in the test set. We split positive predictions ($\sigma(\phi(e_s, r, e_o)) > 0.5$) between (i) *known truths* – training or test/validation instances; and (ii) *other*, the truth of which is not known. We then compute per-relation accuracy over the true training instances ("train") and true test/validation instances ("test"); and the average number of "other" triples predicted true per $e_s, r$ pair. Table 5 shows results for MuRE$_I$, DistMult, TuckER and MuRE. All models achieve near perfect training accuracy. The additive-multiplicative MuRE gives best test set performance, followed (surprisingly) closely by MuRE$_I$, with multiplicative models (DistMult and TuckER) performing poorly on all but type R relations in line with P1(b), with near-zero performance on most type S/C relations. Since the ground truth labels for "other" triples predicted to be true are not in the dataset, we analyse a sample of "other" true predictions for one relation of each type (see Appx. G). From this, we estimate that TuckER is relatively accurate but pessimistic (∼0.3 correct of the 0.5 predictions ≈ 60%), MuRE$_I$ is optimistic but inaccurate (∼2.3 of 7.5 ≈ 31%), whereas MuRE is both optimistic and accurate (∼1.1 of 1.5 ≈ 73%).

**Summary:** Our analysis identifies the best performing model per relation type as predicted by P1(b): multiplicative-only DistMult for type R, additive-multiplicative MuRE for types S/C; providing a basis for *dataset-dependent model selection*. The per-relation insight into where models perform

**Table 5:** Per relation prediction accuracy for MuRE$_I$ (M$_I$), (D)istMult, (T)uckER and (M)uRE (WN18RR).

| Relation Name | Type | $\#_{train}$ | $\#_{test}$ | Accuracy (train) | | | | Accuracy (test) | | | | # Other "True" | | | |
|---|---|---|---|---|---|---|---|---|---|---|---|---|---|---|---|
| | | | | M$_I$ | D | T | M | M$_I$ | D | T | M | M$_I$ | D | T | M |
| verb_group | R | 15 | 39 | 1.00 | 1.00 | 1.00 | 1.00 | 0.97 | 0.97 | 0.97 | 0.97 | 8.3 | 1.7 | 0.9 | 2.7 |
| derivationally_related_form | R | 1714 | 1127 | 1.00 | 1.00 | 1.00 | 1.00 | 0.96 | 0.94 | 0.95 | 0.95 | 8.8 | 0.5 | 0.6 | 1.7 |
| also_see | R | 95 | 61 | 1.00 | 1.00 | 1.00 | 1.00 | 0.64 | 0.64 | 0.61 | 0.59 | 7.9 | 1.6 | 0.9 | 1.9 |
| instance_hypernym | S | 52 | 122 | 1.00 | 1.00 | 1.00 | 1.00 | 0.32 | 0.32 | 0.23 | 0.43 | 1.3 | 0.4 | 0.3 | 0.9 |
| member_of_domain_usage | C | 545 | 43 | 0.98 | 1.00 | 1.00 | 1.00 | 0.02 | 0.00 | 0.02 | 0.00 | 1.5 | 0.6 | 0.0 | 0.3 |
| member_of_domain_region | C | 543 | 42 | 0.88 | 0.89 | 1.00 | 0.93 | 0.02 | 0.02 | 0.00 | 0.02 | 1.0 | 0.4 | 0.8 | 0.7 |
| synset_domain_topic_of | C | 13 | 115 | 1.00 | 1.00 | 1.00 | 1.00 | 0.42 | 0.10 | 0.14 | 0.47 | 0.7 | 0.6 | 0.1 | 0.2 |
| member_meronym | C | 1402 | 307 | 1.00 | 1.00 | 1.00 | 1.00 | 0.22 | 0.02 | 0.01 | 0.22 | 7.9 | 3.4 | 1.5 | 5.6 |
| has_part | C | 848 | 196 | 1.00 | 1.00 | 1.00 | 1.00 | 0.24 | 0.05 | 0.09 | 0.22 | 7.1 | 2.4 | 1.3 | 3.9 |
| hypernym | S | 57 | 1254 | 1.00 | 1.00 | 1.00 | 1.00 | 0.15 | 0.02 | 0.02 | 0.22 | 3.7 | 1.2 | 0.0 | 1.7 |
| all | | 5284 | 3306 | 0.99 | 0.99 | 1.00 | 0.99 | 0.47 | 0.37 | 0.37 | 0.50 | 5.9 | 1.2 | 0.5 | 2.1 |

poorly, e.g. hierarchical or type C relations, can be used to aid and direct future model design. Analysis of the classification performance: (i) shows that MuRE is the most reliable fact prediction model; and (ii) emphasises the poorer ability of multiplicative-only models to represent S/C relations.

### 4.2 P2: PROPERTIES OF RELATION REPRESENTATION

**P2(a)-(b):** Table 6 shows the symmetry score ($\in$ [-1, 1] indicating perfect anti-symmetry to symmetry; see Appx. F) for the relation matrix of TuckER and the norm of relation vectors of TransE, MuRE$_I$ and MuRE on the WN18RR dataset. As expected, type R relations have materially higher symmetry than both other relation types, fitting the prediction of how TuckER compensates for having no additive component. All additive models learn relation vectors of a noticeably lower norm for type R relations, which in the limiting case (similarity) require no additive component, than for types S or C.

**P2(c):** Fig 2 shows eigenvalue magnitudes (scaled relative to the largest and ordered) of relation-specific matrices $\boldsymbol{R}$ of MuRE, labelled by relation type, as predicted to reflect the strength of a relation's *relatedness* aspect. As expected, values are highest for type R relations. For relation types S and C the profiles are more varied, supporting the understanding that relatedness of such relations is highly variable, both in its nature (components of $\mathcal{S}$) and strength (cardinality of $\mathcal{S}$).

**Table 6:** Relation matrix symmetry score [-1.1] for TuckER; and relation vector norm for TransE, MuRE$_I$ and MuRE (WN18RR).

| Relation | Type | Symmetry Score TuckER | Vector Norm | | |
|---|---|---|---|---|---|
| | | | TransE | MuRE$_I$ | MuRE |
| verb_group | R | 0.52 | 5.65 | 0.76 | 0.89 |
| derivationally_related_form | R | 0.54 | 2.98 | 0.45 | 0.69 |
| also_see | R | 0.50 | 7.20 | 0.97 | 0.97 |
| instance_hypernym | S | 0.13 | 18.26 | 2.98 | 1.88 |
| member_of_domain_usage | C | 0.10 | 11.24 | 3.18 | 1.88 |
| member_of_domain_region | C | 0.06 | 12.52 | 3.07 | 2.11 |
| synset_domain_topic_of | C | 0.12 | 23.29 | 2.65 | 1.52 |
| member_meronym | C | 0.12 | 4.97 | 1.91 | 1.97 |
| has_part | C | 0.13 | 6.44 | 1.69 | 1.25 |
| hypernym | S | 0.04 | 9.64 | 1.53 | 1.03 |

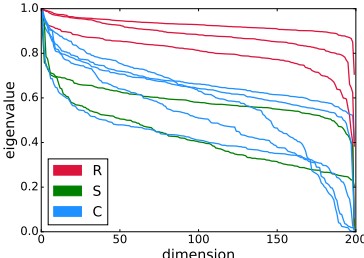

**Figure 2:** Eigenvalue magnitudes of relation-specific matrices $\boldsymbol{R}$ for MuRE by relation type (WN18RR).

## 5 CONCLUSION

Many low-rank knowledge graph representation models have been developed, yet little is known of the latent structure they learn. We build on recent understanding of PMI-based word embeddings to theoretically establish a set of geometric properties of relation representations (relation conditions) required to map PMI-based word embeddings of subject entities to related object entities under knowledge graph relations. These conditions partition relations into three types and provide a basis to consider the loss functions of existing knowledge graph models. Models that satisfy the relation conditions of a particular type have a known set of model parameters that minimise the loss function, i.e. the parameters of PMI embeddings, together with potentially many equivalent solutions. We show that the better a model's architecture satisfies a relation's conditions, the better its performance at link prediction, evaluated under both rank-based metrics and accuracy. Overall, we generalise recent theoretical understanding of how particular semantic relations, e.g. similarity and analogy, are encoded between PMI-based word embeddings to the general relations of knowledge graphs. In doing so, we provide evidence in support of our initial premise: that common latent structure is exploited by both PMI-based word embeddings (e.g. W2V) and knowledge graph representation.

ACKNOWLEDGEMENTS

Carl Allen and Ivana Balažević were supported by the Centre for Doctoral Training in Data Science, funded by EPSRC (grant EP/L016427/1) and the University of Edinburgh.

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

## A  CATEGORISING WN18RR RELATIONS

Table 7 describes how each WN18RR relation was assigned to its respective category.

**Table 7:** Explanation for the WN18RR relation category assignment.

| Type | Relation | Relatedness | Subject Specifics | Object Specifics |
|---|---|---|---|---|
| R | verb_group
derivationally_related_form
also_see | both verbs; similar in meaning
different syntactic categories; semantically related
semantically similar | -
-
- | -
-
- |
| S | hypernym
instance_hypernym | semantically similar
semantically similar | instance of the object
instance of the object | -
- |
| C | member_of_domain_usage
member_of_domain_region
member_meronym
has_part
synset_domain_topic_of | loosely semantically related (word usage features)
loosely semantically related (regional features)
semantically related
semantically related
semantically related | usage descriptor
region descriptor
collection of objects
collection of objects
broad feature set | broad feature set
broad feature set
part of the subject
part of the subject
domain descriptor |

## B  CATEGORISING NELL-995 RELATIONS

Categorisation of NELL-995 relations and the explanation for the category assignment of are shown in Tables 8 and 9 respectively.

**Table 8:** Categorisation of NELL-995 relations.

| Type | Relation | Examples *(subject entity, object entity)* |
|---|---|---|
| R | teamplaysagainstteam
clothingtogowithclothing
agentcollaborateswithagent | *(rangers, mariners), (phillies, tampa_bay_rays)*
*(shirts, trousers), (shoes, black_shirt)*
*(white_stripes, jack_white), (barack_obama, hillary_clinton)* |
| S | professionistypeofprofession
animalistypeofanimal
chemicalistypeofchemical | *(trial_lawyers, attorneys), (engineers, experts)*
*(cats, small_animals), (chickens, livestock)*
*(moisture, gas), (oxide, materials)* |
| C | athleteplayssport
itemfoundinroom
bodypartcontainsbodypart
atdate
locationlocatedwithinlocation
atlocation | *(joe_smith, baseball), (chris_cooley, football)*
*(bed, den), (refrigerator, kitchen_area)*
*(system002, eyes), (blood, left_ventricle)*
*(scotland, n2009), (wto, n2003)*
*(medellin, colombia), (jackson, wyoming)*
*(ogunquin, maine), (palmer_lake, colorado)* |

**Table 9:** Explanation for the NELL-995 relation category assignment.

| Type | Relation | Relatedness | Subject Specifics | Object Specifics |
|---|---|---|---|---|
| R | teamplaysagainstteam
clothingtogowithclothing
agentcollaborateswithagent | both sport teams
both items of clothing that go together
both people or companies; related industries | -
-
- | -
-
- |
| S | professionistypeofprofession
animalistypeofanimal
chemicalistypeofchemical | semantically related (both profession types)
semantically related (both animals)
semantically related (both chemicals) | instance of the object
instance of the object
instance of the object | -
-
- |
| C | athleteplayssport
itemfoundinroom
bodypartcontainsbodypart
atdate
locationlocatedwithinlocation
atlocation | semantically related (sports features)
semantically related (room/furniture features)
emantically related (specific body part features)
loosely semantically related (start date features)
semantically related (geographical features)
semantically related (geographical features) | athlete descriptor
item descriptor
collection of objects
broad feature set
part of the subject
part of the subject | sport descriptor
room descriptor
part of the subject
date descriptor
collection of objects
collection of objects |

## C  SPLITTING THE NELL-995 DATASET

The test set of NELL-995 created by Xiong et al. (2017) contains only 10 out of 200 relations present in the training set. To ensure a fair representation of all training set relations in the validation and test sets, we create new validation and test set splits by combining the initial validation and test sets with the training set and randomly selecting 10,000 triples each from the combined dataset.

## D   IMPLEMENTATION DETAILS

All algorithms are re-implemented in PyTorch with the Adam optimizer (Kingma & Ba, 2015) that minimises binary cross-entropy loss, using hyper-parameters that work well for all models (learning rate: 0.001, batch size: 128, number of negative samples: 50). Entity and relation embedding dimensionality is set to $d_e = d_r = 200$ for all models except TuckER, for which $d_r = 30$ (Balažević et al., 2019b).

## E   KRACKHARDT HIERARCHY SCORE

The Krackhardt hierarchy score measures the proportion of node pairs $(x, y)$ where there exists a directed path $x \to y$, but not $y \to x$; and it takes a value of one for all directed acyclic graphs, and zero for cycles and cliques (Krackhardt, 2014; Balažević et al., 2019a).

Let $M \in \mathbb{R}^{n \times n}$ be the binary *reachability matrix* of a directed graph $\mathcal{G}$ with $n$ nodes, with $M_{i,j} = 1$ if there exists a directed path from node $i$ to node $j$ and 0 otherwise. The Krackhardt hierarchy score of $\mathcal{G}$ is defined as:

$$\text{Khs}_{\mathcal{G}} = \frac{\sum_{i=1}^{n} \sum_{j=1}^{n} \mathbb{1}(M_{i,j} == 1 \wedge M_{j,i} == 0)}{\sum_{i=1}^{n} \sum_{j=1}^{n} \mathbb{1}(M_{i,j} == 1)}. \tag{1}$$

## F   SYMMETRY SCORE

The symmetry score $\in [-1, 1]$ (Hubert & Baker, 1979) for a relation matrix $R \in \mathbb{R}^{d_e \times d_e}$ is defined as:

$$s = \frac{\sum \sum_{i \neq j} R_{ij} R_{ji} - \frac{(\sum \sum_{i \neq j} R_{ij})^2}{d_e(d_e - 1)}}{\sum \sum_{i \neq j} R_{ij}^2 - \frac{(\sum \sum_{i \neq j} R_{ij})^2}{d_e(d_e - 1)}}, \tag{2}$$

where 1 indicates a symmetric and -1 an anti-symmetric matrix.

## G   "OTHER" PREDICTED FACTS

Tables 10 to 13 shows a sample of the unknown triples (i.e. those formed using the WN18RR entities and relations, but not present in the dataset) for the *derivationally_related form* (R), *instance_hypernym* (S) and *synset_domain_topic_of* (C) relations at a range of probabilities ($\sigma(\phi(e_s, r, e_o)) \approx \{0.4, 0.6, 0.8, 1\}$), as predicted by each model. True triples are indicated in bold; instances where a model predicts an entity is related to itself are indicated in blue.

**Table 10:** "Other" facts as predicted by MuRE$_I$.

| Relation (Type) | $\sigma(\phi(e_s, r, e_o)) \approx 0.4$ | $\sigma(\phi(e_s, r, e_o)) \approx 0.6$ | $\sigma(\phi(e_s, r, e_o)) \approx 0.8$ | $\sigma(\phi(e_s, r, e_o)) \approx 1$ |
|---|---|---|---|---|
| derivationally_related_form (R) | (equalizer_NN_2, set_off_VB_5)
(constellation_NN_2, satellite_NN_3)
(**shrink_VB_3, subtraction_NN_2**)
(continue_VB_10, proceed_VB_1)
(support_VB_6, defend_VB_5)
(shutter_NN_1, fill_up_VB_3)
(yawning_NN_1, patellar_reflex_NN_1)
(**yaw_NN_1, spiral_VB_1**)
(stratum_NN_2, social_group_NN_1)
(duel_VB_1, scuffle_NN_3) | (extrapolation_NN_1, maths_NN_1)
(spread_VB_5, circularize_VB_3)
(flaunt_NN_1, showing_NN_2)
(**extrapolate_VB_3, synthesis_NN_3**)
(strategist_NN_1, machination_NN_1)
(crush_VB_4, grind_VB_2)
(spike_VB_5, steady_VB_2)
(licking_NN_1, vanquish_VB_1)
(**synthetical_JJ_1, synthesizer_NN_2**)
(realization_NN_2, embodiment_NN_3) | (sewer_NN_2, stitcher_NN_1)
(lard_VB_1, vegetable_oil_NN_1)
(**snuggle_NN_1, draw_close_VB_3**)
(**train_VB_3, training_NN_1**)
(**scratch_VB_3, skin_sensation_NN_1**)
(scheme_NN_5, schematization_NN_1)
(ordain_VB_3, vest_VB_1)
(lie_VB_1, front_end_NN_1)
(tread_NN_1, step_NN_9)
(**register_NN_3, file_away_VB_1**) | (trail_VB_2, trail_VB_2)
(worship_VB_1, worship_VB_1)
(steer_VB_1, steer_VB_1)
(sort_out_VB_1, sort_out_VB_1)
(make_full_VB_1, make_full_VB_1)
(utilize_VB_1, utilize_VB_1)
(geology_NN_1, geology_NN_1)
(zoology_NN_2, zoology_NN_2)
(uranology_NN_1, uranology_NN_1)
(travel_VB_1, travel_VB_1) |
| instance_hypernym (S) | (thomas_aquinas_NN_1, martyr_NN_2)
(volcano_islands_NN_1, volcano_NN_2)
(cape_horn_NN_1, urban_center_NN_1)
(bergen_NN_1, national_capital_NN_1)
(marshall_NN_2, generalship_NN_1)
(**nansen_NN_1, venturer_NN_2**)
(wisconsin_NN_2, state_capital_NN_1)
(prussia_NN_1, stockade_NN_2)
(**de_mille_NN_1, dancing-master_NN_1**)
(aegean_sea_NN_1, aegean_island_NN_1) | (**taiwan_NN_1, asian_nation_NN_1**)
(**st_gregory_of_n._NN_1, canonization_NN_1**)
(st_gregory_of_n._NN_1, saint_VB_2)
(mccormick_NN_1, find_VB_8)
(**st_gregory_i_NN_1, bishop_NN_1**)
(richard_buckminster_f._NN_1, technological_JJ_2)
(thomas_aquinas_NN_1, archbishop_NN_1)
(**marshall_NN_2, general_officer_NN_1**)
(newman_NN_2, primateship_NN_1)
(thomas_the_apostle_NN_1, sanctify_VB_1) | (**prophets_NN_1, gospels_NN_1**)
(malcolm_x_NN_1, passive_resister_NN_1)
(taiwan_NN_1, national_capital_NN_1)
(truth_NN_5, abolitionism_NN_1)
(**thomas_aquinas_NN_1, saint_VB_2**)
(central_america_NN_1, s._am._nation_NN_1)
(de_mille_NN_1, dance_VB_1)
(st_gregory_i_NN_1, apostle_NN_3)
(fertile_crescent_NN_1, asian_nation_NN_1)
(robert_owen_NN_1, industry_NN_1) | (**helsinki_NN_1, urban_center_NN_1**)
(mannheim_NN_1, stockade_NN_2)
(**nippon_NN_1, nippon_NN_1**)
(victor_hugo_NN_1, novel_NN_1)
(regiomontanus_NN_1, uranology_NN_1)
(**prophets_NN_1, book_NN_10**)
(thomas_aquinas_NN_1, church_father_NN_1)
(woody_guthrie_NN_1, minstrel_VB_1)
(central_america_NN_1, c._am._nation_NN_1)
(aegean_sea_NN_1, island_NN_1) |
| synset_domain_topic_of (C) | (write_VB_8, tape_VB_3)
(introvert_NN_1, scientific_discipline_NN_1)
(**libel_NN_1, slur_NN_2**)
(etymologizing_NN_1, law_NN_1)
(**temple_NN_4, place_of_worship_NN_1**)
(trial_impression_NN_1, proof_VB_1)
(friend_of_the_court_NN_1, war_machine_NN_1)
(**multiv._analysis_NN_1, applied_math_NN_1**)
(**sell_VB_1, transaction_NN_1**)
(draw_VB_6, represent_VB_9) | (draw_VB_6, creative_person_NN_1)
(**suborder_NN_1, taxonomic_group_NN_1**)
(draw_VB_6, draw_VB_6)
(**first_sacker_NN_1, ballgame_NN_2**)
(alchemize_VB_1, modify_VB_3)
(sermon_NN_1, sermon_NN_1)
(**saint_VB_2, catholic_church_NN_1**)
(male_JJ_1, masculine_JJ_2)
(fire_VB_3, zoology_NN_2)
(sell_VB_1, sell_VB_1) | (libel_NN_1, sully_VB_3)
(relationship_NN_4, relationship_NN_4)
(**etymologizing_NN_1, linguistics_NN_1**)
(**turn_VB_12, cultivation_NN_2**)
(brynhild_NN_1, mythologize_VB_2)
(**brynhild_NN_1, myth_NN_1**)
(**assist_NN_2, am._football_game_NN_1**)
(mitzvah_NN_2, human_activity_NN_1)
(drive_NN_12, drive_VB_8)
(**relationship_NN_4, biology_NN_1**) | (**libel_NN_1, disparagement_NN_1**)
(**roll-on_roll-off_NN_1, transport_NN_1**)
(**prance_VB_4, equestrian_sport_NN_1**)
(**libel_NN_1, traducement_NN_1**)
(**sell_VB_1, selling_NN_1**)
(trot_VB_2, ride_horseback_VB_1)
(prance_VB_4, ride_horseback_VB_1)
(gallop_VB_1, ride_horseback_VB_1)
(**brynhild_NN_1, mythology_NN_2**)
(**drive_NN_12, badminton_NN_1**) |

**Table 11:** "Other" facts as predicted by DistMult.

| Relation (Type) | $\sigma(\phi(e_s, r, e_o)) \approx 0.4$ | $\sigma(\phi(e_s, r, e_o)) \approx 0.6$ | $\sigma(\phi(e_s, r, e_o)) \approx 0.8$ | $\sigma(\phi(e_s, r, e_o)) \approx 1$ |
|---|---|---|---|---|
| derivationally_ related_form (R) | (stag_VB_3, undercover_work_NN_1) (print_VB_4, publisher_NN_2) (crier_NN_3, pitchman_NN_2) (play_VB_26, turn_NN_10) (count_VB_4, recite_VB_2) (vividness_NN_2, imbue_VB_3) (sea_mew_NN_1, larus_NN_1) (alkali_NN_2, acidify_VB_2) (see_VB_17, understand_VB_2) (shun_VB_1, hedging_NN_2) | (dish_NN_2, stew_NN_2) (expose_VB_3, show_NN_1) (system_NN_9, orderliness_NN_1) (spread_NN_4, strew_VB_1) (take_down_VB_2, put_VB_2) (wrestle_VB_4, wrestler_NN_1) (autotr._organism_NN_1, epiphytic_JJ_1) (duel_VB_1, slugfest_NN_1) (vocal_NN_2, rock_star_NN_1) (smelling_NN_1, scent_VB_1) | (shrink_NN_1, pedology_NN_1) (finish_VB_6, finishing_NN_2) (play_VB_26, playing_NN_3) (centralization_NN_1, unite_VB_6) (existence_NN_1, living_NN_3) (mouth_VB_3, sassing_NN_1) (constellation_NN_2, star_NN_1) (print_VB_4, publishing_house_NN_1) (puzzle_VB_2, secret_NN_3) (uranology_NN_1, tt_NN_1) | (alliterate_VB_1, versifier_NN_1) (geology_NN_1, structural_JJ_5) (resect_VB_1, amputation_NN_2) (nutrition_NN_3, man_NN_4) (saint_NN_3, sanctify_VB_1) (right_fielder_NN_1, leffield_NN_1) (list_VB_4, slope_NN_2) (lieutenancy_NN_1, captain_NN_1) (tread_NN_1, step_VB_7) (exenteration_NN_1, enucleate_VB_2) |
| instance_ hypernym (S) | (wisconsin_NN_2, urban_center_NN_1) (marshall_NN_2, lieutenant_general_NN_1) (abidjan_NN_1, cote_d'ivoire_NN_1) (world_war_i_NN_1, urban_center_NN_1) (st._paul_NN_2, evangelist_NN_2) (deep_south_NN_1, urban_center_NN_1) (nuptse_NN_1, urban_center_NN_1) (ticino_NN_1, urban_center_NN_1) (aegean_sea_NN_1, aegean_island_NN_1) (cowpens_NN_1, war_of_am._ind._NN_1) | (mississippi_river_NN_1, american_state_NN_1) (r._e._byrd_NN_1, commissioned_officer_NN_1) (kobenhavn_NN_1, urban_center_NN_1) (the_gambia_NN_1, africa_NN_1) (tirich_mir_NN_1, urban_center_NN_1) (r._e._byrd_NN_1, military_advisor_NN_1) (r._e._byrd_NN_1, aide-de-camp_NN_1) (tampa_bay_NN_1, urban_center_NN_1) (tidewater_region_NN_1, south_NN_1) (r._e._byrd_NN_1, executive_officer_NN_1) | (deep_south_NN_1, south_NN_1) (capital_of_gambia_NN_1, urban_center_NN_1) (south_west_africa_NN_1, africa_NN_1) (brandenburg_NN_1, urban_center_NN_1) (sierra_nevada_NN_1, urban_center_NN_1) (malcolm_x_NN_1, emancipationist_NN_1) (north_platte_river_NN_1, urban_center_NN_1) (oslo_NN_1, urban_center_NN_1) (zaire_river_NN_1, urban_center_NN_1) (transylvanian_alps_NN_1, urban_center_NN_1) | (helsinki_NN_1, urban_center_NN_1) (the_nazarene_NN_1, save_VB_7) (irish_capital_NN_1, urban_center_NN_1) (r._e._byrd_NN_1, inspector_general_NN_1) (r._e._byrd_NN_1, chief_of_staff_NN_1) (central_america_NN_1, c._am._nation_NN_1) (malcolm_x_NN_1, environmentalist_NN_1) (the_nazarene_NN_1, christian_JJ_1) (thomas_aquinas_NN_1, church_father_NN_1) (the_nazarene_NN_1, el_nino_NN_2) |
| synset_domain_ topic_of (C) | (limitation_NN_4, trammel_VB_2) (light_colonel_NN_1, colonel_NN_1) (nurse_VB_1, nursing_NN_1) (sermon_NN_1, prophesy_VB_2) (libel_NN_1, practice_of_law_NN_1) (slugger_NN_1, baseball_player_NN_1) (rna_NN_1, chemistry_NN_1) (metrify_VB_1, versify_VB_1) (trial_impression_NN_1, publish_VB_1) (turn_VB_12, plowman_NN_1) | (roll-on_roll-off_NN_1, transport_NN_1) (hizb_ut-tahrir_NN_1, asia_NN_1) (slugger_NN_1, softball_game_NN_1) (sermon_NN_1, sermonize_VB_1) (draw_VB_6, drawer_NN_3) (turn_VB_12, plow_NN_1) (assist_NN_2, softball_game_NN_1) (council_NN_2, assembly_NN_4) (throughput_NN_1, turnout_NN_4) (cream_VB_1, cream_NN_2) | (etymologizing_NN_1, explanation_NN_1) (ferry_VB_3, travel_VB_1) (public_prosecutor_NN_1, prosecute_VB_2) (alchemize_VB_1, modify_VB_3) (libel_NN_1, libel_VB_1) (turn_VB_12, till_VB_1) (hit_NN_1, hit_VB_1) (fire_VB_3, flaming_NN_1) (ring_NN_4, chemical_chain_NN_1) (libidinal_energy_NN_1, charge_NN_9) | (flat_JJ_5, matte_NN_2) (etymologizing_NN_1, derive_VB_3) (hole_out_VB_1, hole_NN_3) (relationship_NN_4, relation_NN_1) (drive_NN_12, badminton_NN_1) (etymologizing_NN_1, etymologize_VB_2) (matrix_algebra_NN_1, diagonalization_NN_1) (cabinetwork_NN_2, woodworking_NN_1) (cabinetwork_NN_2, bottom_VB_1) (cabinetwork_NN_2, upholster_VB_1) |

**Table 12:** "Other" facts as predicted by TuckER.

| Relation (Type) | $\sigma(\phi(e_s, r, e_o)) \approx 0.4$ | $\sigma(\phi(e_s, r, e_o)) \approx 0.6$ | $\sigma(\phi(e_s, r, e_o)) \approx 0.8$ | $\sigma(\phi(e_s, r, e_o)) \approx 1$ |
|---|---|---|---|---|
| derivationally_related_form (R) | (tympanist_NN_1, gong_NN_2)
(indication_NN_1, signalize_VB_2)
(turn_over_VB_3, rotation_NN_3)
(date_VB_5, geological_dating_NN_1)
(set_VB_23, emblem_NN_2)
(tyro_NN_1, start_VB_5)
(identification_NN_1, name_VB_5)
(stabber_NN_1, thrust_VB_5)
(justification_NN_1, apology_NN_2)
(manufacture_VB_1, prevarication_NN_1) | (mash_NN_2, mill_VB_2)
(walk_VB_9, zimmer_frame_NN_1)
(use_VB_5, utility_NN_2)
(musical_instrument_NN_1, write_VB_6)
(lining_NN_3, wrap_up_VB_1)
(scrap_VB_2, struggle_NN_2)
(tape_VB_3, tape_recorder_NN_1)
(vindicate_VB_2, justification_NN_2)
(leaching_NN_1, percolate_VB_3)
(synchronize_VB_2, synchroscope_NN_1) | (take_chances_VB_1, venture_NN_1)
(shutter_NN_1, fill_up_VB_3)
(exit_NN_3, leave_VB_1)
(trembler_NN_1, vibrate_VB_1)
(motivator_NN_1, trip_VB_4)
(support_VB_6, indorsement_NN_1)
(federate_VB_2, confederation_NN_1)
(take_over_VB_6, return_NN_7)
(wait_on_VB_1, supporter_NN_3)
(denote_VB_3, promulgation_NN_1) | (venturer_NN_2, venturer_NN_2)
(dynamitist_NN_1, dynamitist_NN_1)
(love_VB_3, lover_NN_2)
(snuggle_NN_1, squeeze_VB_8)
(departed_NN_1, die_VB_2)
(position_VB_1, placement_NN_1)
(repentant_JJ_1, repentant_JJ_1)
(tread_NN_1, step_VB_7)
(stockist_NN_1, stockist_NN_1)
(philanthropist_NN_1, philanthropist_NN_1) |
| instance_hypernym (S) | (deep_south_NN_1, south_NN_1)
(st_paul_NN_2, organist_NN_1)
(helsinki_NN_1, urban_center_NN_1)
(malcolm_x_NN_1, emancipationist_NN_1)
(thomas_the_apostle_NN_1, church_father_NN_1)
(st_gregory_of_n._NN_1, sermonizer_NN_1)
(robert_owen_NN_1, movie_maker_NN_1)
(theresa_NN_1, monk_NN_1)
(st_paul_NN_2, philosopher_NN_1)
(ibn-roshd_NN_1, pedagogue_NN_1) | (thomas_aquinas_NN_1, bishop_NN_1)
(irish_capital_NN_1, urban_center_NN_1)
(thomas_the_apostle_NN_1, apostle_NN_2)
(st_paul_NN_2, apostle_NN_3)
(mccormick_NN_1, painter_NN_1)
(thomas_the_apostle_NN_1, troglodyte_NN_1)
(mccormick_NN_1, electrical_engineer_NN_1)
(mississippi_river_NN_1, american_state_NN_1) | (cowpens_NN_1, siege_NN_1)
(mccormick_NN_1, arms_manufacturer_NN_1)
(thomas_the_apostle_NN_1, evangelist_NN_2)
(mccormick_NN_1, technologist_NN_1)
(st_gregory_i_NN_1, church_father_NN_1) | (r._e._byrd_NN_1, siege_NN_1)
(shaw_NN_3, women's_rightist_NN_1)
(aegean_sea_NN_1, aegean_island_NN_1)
(thomas_aquinas_NN_1, church_father_NN_1) |
| synset_domain_topic_of (C) | (roll-on_roll-off_NN_1, motorcar_NN_1)
(libel_NN_1, legislature_NN_1)
(roll-on_roll-off_NN_1, passenger_vehicle_NN_1) | (drive_NN_12, badminton_NN_1) | | |

**Table 13:** "Other" facts as predicted by MuRE.

| Relation (Type) | $\sigma(\phi(e_s, r, e_o)) \approx 0.4$ | $\sigma(\phi(e_s, r, e_o)) \approx 0.6$ | $\sigma(\phi(e_s, r, e_o)) \approx 0.8$ | $\sigma(\phi(e_s, r, e_o)) \approx 1$ |
|---|---|---|---|---|
| derivationally_related_form (R) | (surround_VB_1, wall_NN_1)
(unpleasant_JJ_1, unpalatableness_NN_1)
(love_VB_3, enjoyment_NN_2)
(magnitude_NN_1, tall_JJ_1)
(testify_VB_2, information_NN_1)
(connect_VB_6, converging_NN_1)
(connect_VB_6, connexion_NN_4)
(operate_VB_4, psyop_NN_1)
(market_VB_1, trade_NN_4)
(operate_VB_4, mission_NN_2) | (word_picture_NN_1, sketch_VB_2)
(develop_VB_10, adjustment_NN_4)
(gloss_VB_3, commentary_NN_1)
(violate_VB_2, violation_NN_3)
(suffocate_VB_1, strangler_tree_NN_1)
(number_VB_3, point_NN_12)
(develop_VB_10, organic_process_NN_1)
(plication_NN_1, twist_VB_4)
(split_up_VB_3, separation_NN_5)
(plication_NN_1, wrinkle_VB_2) | (smelling_NN_1, wind_VB_4)
(try_out_VB_1, somatic_cell_nuclear_transplantation_NN_1)
(lighting_NN_4, set_on_fire_VB_1)
(symphalangus_NN_1, one-half_NN_1)
(just_JJ_3, validity_NN_1)
(reprove_VB_1, talking_to_NN_1)
(sustain_VB_5, beam_NN_2)
(spring_NN_6, hurdle_VB_1)
(spark_NN_1, scintillate_VB_1)
(utility_NN_2, functional_JJ_1) | (spoliation_NN_2, sack_VB_1)
(desire_NN_2, hope_VB_2)
(snuffle_VB_3, whine_NN_1)
(nasalization_NN_1, sound_out_VB_1)
(tread_NN_1, step_VB_7)
(yearn_VB_1, pining_NN_1)
(unreliableness_NN_1, arbitrary_JJ_1)
(travesty_NN_2, travesty_NN_2)
(spark_NN_1, sparkle_VB_1)
(stockist_NN_1, stockist_NN_1) |
| instance_hypernym (S) | (malcolm_x_NN_1, hipster_NN_1)
(the_nazarene_NN_1, judaism_NN_2)
(old_line_state_NN_1, river_NN_1)
(r_e_byrd_NN_1, commissioned_officer_NN_1)
(south_korea_NN_1, peninsula_NN_1)
(st_gregory_of_n._NN_1, vicar_of_christ_NN_1)
(nippon_NN_1, italian_region_NN_1)
(robert_owen_NN_1, tycoon_NN_1)
(mandalay_NN_1, national_capital_NN_1)
(nan_ling_NN_1, urban_center_NN_1) | (central_america_NN_1, central_america_NN_1)
(st._gregory_i_NN_1, church_father_NN_1)
(south_korea_NN_1, african_nation_NN_1)
(malcolm_x_NN_1, passive_resister_NN_1)
(malcolm_x_NN_1, birth-control_reformer_NN_1)
(los_angeles_NN_1, port_NN_1)
(great_lakes_NN_1, canadian_province_NN_1)
(transylvanian_alps_NN_1, urban_center_NN_1)
(gettysburg_NN_2, siege_NN_1)
(wisconsin_NN_2, geographical_region_NN_1) | (theresa_NN_1, monk_NN_1)
(nippon_NN_1, european_nation_NN_1)
(great_lakes_NN_1, river_NN_1)
(r._e._byrd_NN_1, noncommissioned_officer_NN_1)
(world_war_i_NN_1, pitched_battle_NN_1)
(irish_capital_NN_1, urban_center_NN_1)
(volcano_islands_NN_1, urban_center_NN_1)
(nippon_NN_1, american_state_NN_1)
(helsinki_NN_1, urban_center_NN_1)
(capital_of_gambia_NN_1, urban_center_NN_1) | |
| synset_domain_topic_of (C) | (libel_NN_1, criminal_law_NN_1)
(brynhild_NN_1, mythology_NN_2)
(slugger_NN_1, sport_NN_1)
(sell_VB_1, law_NN_1)
(semitic_deity_NN_1, mythology_NN_1)
(nuclear_deterrence_NN_1, law_NN_1)
(reception_NN_5, baseball_game_NN_1)
(photosynthesis_NN_1, chemistry_NN_1)
(isolde_NN_1, parable_NN_1)
(assist_NN_2, court_game_NN_1) | (write_VB_8, transcription_NN_5)
(temple_NN_4, muslimism_NN_2)
(assist_NN_2, hockey_NN_1)
(relationship_NN_4, biology_NN_1)
(apostle_NN_3, western_church_NN_1)
(assist_NN_2, sport_NN_1)
(trot_VB_2, equestrian_sport_NN_1)
(rna_NN_1, chemistry_NN_1)
(assist_NN_2, soccer_NN_1)
(assist_NN_2, football_game_NN_1) | (assist_NN_2, am._football_game_NN_1)
(drive_NN_12, court_game_NN_1)
(sell_VB_1, offense_NN_3)
(slugger_NN_1, softball_game_NN_1)
(drive_NN_12, badminton_NN_1) | |