# OpenReview forum: "Interpreting Knowledge Graph Relation Representation from Word Embeddings"
_ICLR.cc/2021/Conference — ICLR 2021 Poster_

### Official Review · AnonReviewer1 · 2020-10-27
**Nothing surprised me.**

**Rating:** 7
**Confidence:** 4

**Review:**

Based on PMI word embedding, the authors categorize the knowledge graph relations into three types, which serve as the foundation of knowledge analysis. This paper is not well-motived but presents the methodology, well. However, nothing in this paper surprised me, because this seems like a ````''regular'' research in this field.

Major Concerns:
1. The motivation is not crystally clear. I suggest the authors explain the motivation in detail, e.g., why different embedding are fundamentally the same (strong claim but no evidence).  I suppose the author would like to align the structure of linguistic semantics and knowledge semantics. However, there is no sufficient evidence for me to accept this alignment. Can you give me some examples or strong evidence to claim the joint learning is not incremental.

2. This paper is indeed fine, except for motivation. Good presentation, the clear methodology, and promising results. However, there are plenty of researches to join the textual info and knowledge info together, which makes this paper not very novel. Besides, based on PMI embedding and relation categorization are still very regular in the field. Thus, I doubt the novelty of this paper. However, compared to these papers published in this field, I think this paper can be accepted. This is the reason why I rate 7 but I still got unsatisfactory about the novelty.

Minor Concerns:
1. Fig.1: Please specify the legend clearly.
2. Table 2: For more beginning readers, please revise the examples to human-readable.

Discussion:

I think the categorization shall benefit the performance most when the difficulties between relation categories are balanced. Does this sound like a correct claim? because in your experiment, I found R is the easiest while S is hardest, and the difficulty different for the two types is large. I don't judge this question. I just want to provide a new idea for your paper to improve.

---

> ### Author Response · Authors · 2020-11-17
> **Response to Reviewer 1**
>
> Many thanks for your review.
>
> **“Motivation”**\
> Our main motivation is to try to understand different KG model performance by developing a theory supported by empirical evidence. Our aim is that this contributes to a theoretical foundation for a largely empirical field, offering a principled direction for KG representation model development. \
> Our specific approach is motivated by recent understanding for PMI-based word embeddings based on word co-occurrences. Words co-occur more than “by chance” (PMI>0) if they are related in some way. As such, semantic relationships induce structure in co-occurrence statistics, which manifest in geometric relationships between PMI-based embeddings of related words, e.g. analogies. We extend this semantics-to-geometry connection (shown previously for general relation types, e.g. analogy, paraphrase, similarity in [1, 2]) to the specific relations of knowledge graphs. For different relation types, we derive geometric properties of relation representations required to map between PMI-based word embeddings (which cannot be done for general “entity embeddings” since we have little understanding of their structure). The derived geometric properties can be considered a (particular) theoretical model of how relations can be represented. We then show that the better a model fits this theoretical model, the better it performs on a relation-by-relation basis. \
> The “motivating premise” for our approach (p.1) makes no strong claim but, since (a) PMI-based word embeddings are observed to capture semantic relationships between words and (b) KG representation models aim to capture semantic relationships between words, we proceed on the basis that an understanding from one might help the other.
>
>
> **“Nothing in this paper surprised me, seems like ''regular'' research”, “Paper not very novel”**\
> Whilst much empirical research exists in both the fields of KG representation and PMI-based word embedding, a theoretical connection between semantics and geometry of word embeddings has only been drawn recently [1, 2]; and we are unaware of a theoretical explanation for properties of KG relation representations, beyond limited specific observations (e.g. asymmetric mappings are required to represent asymmetric relations). We extend the recent theoretical understanding [1, 2] for a limited set of general relation types to develop a model for the geometric relationships between PMI-based word embeddings, corresponding to specific relations of KGs. We then show that the more the loss function of a KG model is compatible with those geometric relationships, the better it performs empirically in modelling those relations. We believe that deriving geometric properties of KG relation representations based on an understanding of how semantics can correspond to geometry is novel. We also believe that our proposed categorisation of relation types based on relation conditions is novel.
>
>
> **“Fig.1: Please specify the legend clearly.”** \
> Thanks for the suggestion, we have updated the legend (which should be read in conjunction with Section 3 text).
>
> **“Table 2: For more beginning readers, please revise the examples to human-readable.”**\
> Table 2 includes examples of relations and subject/object word pairs taken from the WN18RR data_set. We agree with the reviewer that the table could be formatted to be easier to read, but on balance believe that the current presentation gives a transparent link to the dataset (e.g. enabling these instances to be readily found), preserves all information (e.g. part-of-speech tag) and involves no subjective adjustment.
>
> **“I think the categorization shall benefit the performance most when the difficulties between relation categories are balanced. Does this sound like a correct claim?”**\
> The relation categorisation specifies the geometric relationship between word embeddings for each relation type. The geometry corresponding to each relation type is reflective of semantic properties of the relation, e.g. some relations are relatively “vague/loose” (e.g. “member_of_domain_region”: <rome, gladiator>, <USA, multiple_voting>), which is reflected in a less tightly defined geometric relationship. We believe that such “looseness” increases the difficulty for a model to identify certain relations (analogously to the increasing difficulty human annotators might have in identifying increasingly vague relations).
>
> [1] Carl Allen and Timothy Hospedales. Analogies Explained: Towards Understanding Word Embeddings. ICML, 2019.\
> [2] Carl Allen, Ivana Balazevic, and Timothy Hospedales. What the Vec? Towards Probabilistically Grounded Embeddings. NeurIPS, 2019.

---

### Official Review · AnonReviewer3 · 2020-10-28
**Interpreting Knowledge Graph Relation Representation From Word Embeddings**

**Rating:** 7
**Confidence:** 3

**Review:**

Summary and Contributions

The authors study the latent semantic properties of word representation models by categorising relations between entities.  The goal is to show that word embeddings and knowledge graph representations learn a common latent structure even if both types of models have different learning objectives.  The main contributions are the mapping of relations between subjects to object word embeddings, categorisation of such relations, and evaluation of the state-of-the-art knowledge graph representations. The study shows that knowledge representation models follow the defined relation conditions.

Strengths

- Clear description of background knowledge needed to understand the proposed study.
- The authors perform a comprehensive comparison across different knowledge graph representations.
- The findings show that there is a connection in the lower dimensional space between word and graph representations.

Weaknesses

- It is not clear which particular framework is used to define the hierarchy of knowledge graph relations.
- It is not discussed the possible benefit of the learned latent structure of knowledge graph models for the performance on downstream tasks, e.g.  text classification, or natural language inference.

Questions to the Authors

- How the main contributions relate to the experiments P1 and P2?
- Could you elaborate on how you define the hierarchy knowledge graph relations.
- Which is the relation between the used knowledge graph categorisation and the related work?

---

> ### Author Response · Authors · 2020-11-17
> **Response to Reviewer 3**
>
> Many thanks for your review.
>
> **“It is not clear which particular framework is used to define the hierarchy of knowledge graph relations.”**\
> Where a relation induces a hierarchical (tree) structure over entities, the “type” of that relation is either specialisation (S) or context shift (C) (by definition, hierarchies are asymmetric so hierarchical relations cannot be type R). For example, the relation “part of” induces a hierarchy, e.g. bus, wheel, tyre, where words share a common semantic theme (implying a “relatedness” aspect), but are not instances of each other and so the relation is a (generalised) context shift (C) type. The relation “instance of” forms another hierarchy, e.g. person, officer, petty officer. Words again share a common semantic theme (implying “relatedness”) but are specialisations of each other, so fall under the (generalised) specialisation (S) type.
> To try to identify hierarchical relations, we use “Krackhardt score” and path length in Tables 3 & 4 (following [1]). However, both of these metrics rely on a KG containing sufficient instances of a relation, since if e.g. a KG contains only “one-way” instances of a symmetric relation (i.e. <a,b> but not <b,a>), then the relation may have a high Krackhardt score/path length and so appear hierarchical even if it is not (e.g. see in Table 3 the relation “also see” capturing synonyms, e.g. <clean, tidy>, <ram, screw>).
>
> **“It is not discussed the possible benefit of the learned latent structure of knowledge graph models for the performance on downstream tasks, e.g. text classification, or natural language inference.”**\
> The aim of this work is to develop a theoretical understanding of how entities and relations can be represented and to bridge the relatively disassociated fields of KG representation and word embedding. Although beyond the scope of our current work, we would hope/expect this to benefit the future development of representation/embedding models and to help interpret downstream processes that use these representations.
>
> **“How the main contributions relate to the experiments P1 and P2?”**\
> Prediction P1 relates to high level properties (e.g. model/relation-specific performance) based on our relation categorisation (R, S, C) and the derived geometric structure of relation representations. The first key contribution (p.2) relates to the theory behind this prediction, the second key contribution (note: we have re-ordered them) corresponds to the experiments that support it.
> Prediction P2 relates to specific properties of relation representations. The third key contribution relates to the experiments that support P2, e.g. looking at symmetry and eigenvalues of the relation matrix and the vector norm of the relation vector.
> Thanks, we have added cross-references to the key contributions for greater clarity.
>
> **“Which is the relation between the used knowledge graph categorisation and the related work?”**\
> Various ways of describing/categorising knowledge graph relations exist e.g. by symmetry/asymmetry (as identified in [2]) and hierarchy (as identified in [1]). The proposed categorisation, which is based on relation conditions, is novel and delineates relations by the required mathematical form (and complexity) of their representation.
>
> [1] Ivana Balazevic, Carl Allen and Timothy M Hospedales. Multi-relational Poincaré Graph Embeddings. NeurIPS 2019.\
> [2] Théo Trouillon, Johannes Welbl, Sebastian Riedel, Éric Gaussier, and Guillaume Bouchard. Complex Embeddings for Simple Link Prediction. ICML, 2016.

---

### Official Review · AnonReviewer2 · 2020-10-28
**This paper contributes to understanding the latent structure of low-rank knowledge graph representations. The authors draw a parallel between the embeddings of knowledge graphs and words, in the sense that they capture the same structure of relations.**

**Rating:** 7
**Confidence:** 3

**Review:**

Recent works toward the understanding of word embeddings can explain how semantic word relationships, such as similarity, analogy and paraphrasing are encoded as low-rank projections of high dimensional vectors of co-occurrence statistics (Allen et al., 2019). Thus, the semantic relationships correspond to linear relationships of word embeddings. This paper builds on this understanding of (PMI-based) word embeddings aiming at the task of understanding the latent structure of low-rank knowledge graph representations. The authors draw a parallel between the embeddings of knowledge graphs and words under the premise that fundamentally the same structure of relations is captured in different ways. Strong evidence to this premise is provided by starting at encoded semantic relations of word embeddings generalizing them to three types (R,S,C) of knowledge graph relations. The authors analyse the performance of different state-of-the-art knowledge graph models and identify the best performing model per relation type. While a multiplicative model performs best for R-relations (highly related), an additive-multiplicative model should be used for S- (specialisation) or C-type (context-shift) relations. These results correspond to the predictions made beforehand and the theoretically derived loss functions based on the respective conditions of each relation type.

Pros:
•	The paper is technically sound, well written and organized and free of typographical errors
•	It focusses on the timely and interesting problem of understanding the latent structure of knowledge graph representations
•	The key strength of the paper is the idea to categorize relations between entities based on the geometric properties of relation representations of word embeddings
•	The theoretically derived loss functions are a strong evidence to why different kind of knowledge graph models perform unequally on the defined relation types

Minor comments:
•	The relation types in the text (four bullet points, p. 4) should perhaps mirror one-to-one the types in figure 1
•	Context-shift relations are explained as "subject to synonyms", a specialisation should perhaps be explained as hypernyms and context shift e.g. as meronyms

---

> ### Author Response · Authors · 2020-11-17
> **Response to Reviewer 2**
>
> Many thanks for your review.
>
> **“The relation types in the text (four bullet points, p. 4) should perhaps mirror one-to-one the types in figure 1”**\
> We have considered this suggestion and agree it would increase clarity in a particular respect. However, we believe that the “specialisation” subfigure is useful in developing an understanding of different relation types. To try to balance these, we have made the word “specialisation” bold.
>
> **“Context-shift relations are explained as "subject to synonyms", a specialisation should perhaps be explained as hypernyms and context shift e.g. as meronyms”**\
> Thanks, we have added this to the paper.

---

### Official Review · AnonReviewer4 · 2020-10-29
**Clear exposition and hypothesis but underwhelming empirical validation**

**Rating:** 6
**Confidence:** 3

**Review:**

This paper aims to establish a theoretical basis for geometric properties of knowledge graph relations and embedded entities by comparing knowledge graph embeddings with word embeddings. Using the insight that the semantic properties of PMI-based word embeddings manifest as linear geometric relationships, they view and compare the relationship embeddings derived from different knowledge graph embedding schemes in this way.
The analysis claims to show that when the KG architecture conforms to the presented relation types and conditions (divided into similarity, relatedness, and context shift types), it has better performance of link prediction for that embedding scheme.

The empirical evaluation focuses on comparison of 4 embedding schemes that have linear transformation score functions (additive, multiplicative, both) on WN18RR and NELL-995 relations for link prediction on several examples of the relation types.

Overall, the paper is well-motivated, cites relevant literature in the theory behind word embeddings, and is generally clearly written. It has a useful proposal for the types and conditions of the three relation types and clear hypothesis for the performance of knowledge graph relation transformations that have certain properties.

However, the empirical evaluation does not seem to completely support the claims. TransE is obviously lower performing across relations, but MureI seems quite close in most cases to models that involve multiplacative relatedness, so it’s not obvious to me that MureI performs worse. Further, the summery suggests that DistMult is preferable for type R, but MuRe appears to do equally well or better on most cases, thus it’s not clear under what circumstances (what dataset dependent factor) would point to not choosing MuRe.
I would expect to see a starker contrast between the performance of the different models per claim type to support the dataset dependent statement. Perhaps another experimental setting, like comparison on non-linear transformation, or other examples, would help support that claim.

---

> ### Author Response · Authors · 2020-11-17
> **Response to Reviewer 4**
>
> Many thanks for your review.
>
> **“MureI seems quite close in most cases to models that involve multiplicative relatedness”**\
> We predict that representations of S and C type relations can require both additive and multiplicative components. Based on comparison of TransE and DistMult (neither of which contain biases and so are fairly comparable), we mention that additive-only models appear in general to perform worse than multiplicative-only. However, the main point is that both additive and multiplicative components are preferable, as shown by the performance of MuRE particularly for S/C relations.
>
> **“the summary suggests that DistMult is preferable for type R, but MuRe appears to do equally well or better … it’s not clear under what circumstances (what dataset dependent factor) would point to not choosing MuRe.”**\
> That observation is correct: certainly MuRE is the best performing model overall since it is most able to represent all relation types. Our point regarding model selection is that if it were known that a dataset contained only type R relations, then DistMult may be a suitable model choice since the extra additive component of MuRE (useful for other relation types) would be superfluous. However, such redundancy of the additive component would need to be learned or else be detrimental to performance. This is supported in Table 4 for the NELL dataset, where DistMult outperforms all other models for type R relations (team_plays_against_team, clothing_to_go_with_clothing and againt_collaborates_with_agent).

---

### Decision · Program_Chairs · 2021-01-07
**Final Decision**

**Decision:**

Accept (Poster)

**Comment:**

This paper extends the recent theoretical understanding on geometric properties for word embeddings to relations and entities of knowledge graph. It categorizes relations into different types and derive requirements for their representations. Empirically they experiment several graph embedding approaches and show that when the loss function is aligned with the requirement of the relation type, we can achieve better performance.  The reviewers generally find the paper to be solid, well executed and provides useful insights. The authors are encouraged to strengthen the discussion of the motivation of this work, and improve the presentation based on reviewers' comments.